# A French Adaptation and Validation of Retirement Semantic Differential (RSD)

**DOI:** 10.3390/bs14100891

**Published:** 2024-10-02

**Authors:** Laurie Borel, Benjamin Boller, Georg Henning, Guillaume T. Vallet

**Affiliations:** 1Département de Psychologie, Université du Québec à Trois-Rivières, Trois-Rivières, QC G8Z 4M3, Canada; laurie.borel@uqtr.ca (L.B.); benjamin.boller@uqtr.ca (B.B.); 2Unité de Formation et de Recherche en Psychologie, Université Clermont Auvergne, CNRS, LaPSCo, F-63000 Clermont–Ferrand, France; 3Centre de Recherche de l’Institut Universitaire de Gériatrie de Montréal, Montreal, QC H3W 1W5, Canada; 4German Centre of Gerontology, 12101 Berlin, Germany; georg.henning@dza.de

**Keywords:** retirement, attitudes, social representation, scale, health

## Abstract

Retirement is associated with numerous representations, some of them being negative and the other positive. Yet, these representations affect the health of individuals in their transition to retirement. However, although the socio-political context in France favors the emergence of numerous representations of retired people, to our knowledge there is no scale validated in French that would allow us to evaluate them. Thus, the objective of this study was to adapt and validate a scale assessing representations of retired people, called the Retirement Semantic Differential (RSD), for a French population. The scale consists of a series of bipolar adjectives related to retirement, such as “active/passive” and “happy/sad”, with participants’ responses indicating the connotative meaning, positive or negative, about representations of retirement. A total of 279 participants aged 18 to 55 years, recruited online, completed the adapted version of the RSD. The results show that the scale has good psychometric properties. The analysis found a three-factor model, and some items were removed, resulting in a reduced version of the scale (11 items). The results will be discussed in terms of cultural and socio-political differences. This scale could contribute to a better understanding of the deleterious effects on health of the transition to retirement and serve to improve the effectiveness of interventions aimed at reducing the negative effects of these representations upon young retirees or those preparing for retirement.

## 1. Introduction

Ageing well is a major societal issue and is therefore the subject of growing interest, particularly considering the rapid increase in the proportion of older people and retired people in our societies. Nonetheless, ageing and retirement are associated with numerous negative social representations, which can, in the long term, have deleterious repercussions on the physical and mental health of older adults [1]. Whereas numerous studies have focused on the representation of ageing [2] very little is known about the social representations of retirement. Retirement constitutes a crucial stage of transitions, a source of many changes, in an individual’s life [3]. As such, some representations of retirement may be different and specific from those on ageing, especially just after retirement. Among the few studies conducted on the effect of the representation of retirement, it has been found that negative representations negatively impact health, such as longevity [1,4]. However, there is currently no validated scale in French that can specifically evaluate these representations and subsequently study their effects on health. The objective of this study is to adapt and validate a scale assessing representations of retired people, called the Retirement Semantic Differential (RSD), for a French population [1]. A total of 279 participants aged 18 to 55 years completed the adapted version of the RSD online by answering 14 polarized propositions about a retired person (e.g., independent/dependent) on a 7-point Likert scale. The results will be discussed in terms of cultural and socio-political differences. The validation of this scale could contribute to a better understanding of the deleterious health effects of the transition to retirement and serve to improve the effectiveness of interventions aimed at reducing the negative effects of representations upon young retirees or those preparing for retirement.

“I can’t wait for retirement!” is probably one of the sentences that each worker has said in his/her career more than once. Retirement is often perceived as a well-deserved time of rest, generating less stress and allowing more daily freedom [5]. Indeed, the cessation of professional activity is synonymous with many changes and constitutes a major transition for individuals, affecting their daily, social and emotional life [6]. Model of retirement adjustment [7], predicts that attitudes toward retirement change over time, especially in the months following retirement. These changes can influence the evolution of health after retirement and depend on numerous factors. Indeed, even if the effects of the passage to retirement on mental and physical health are mixed, there is great inter-individual heterogeneity in the evolution of health after retirement [8]. While some factors appear to have a protective effect, such as physical activity [9] and social support [10], retirement appears to be associated with negative effects for some groups, or for some outcomes, such as mandatory or involuntary retirement [11].

However, while some people look forward to retirement, it also generates uncertainties [12], and it appears that people’s health status may deteriorate shortly after retirement [13]. In France, although life expectancy is around 80 years for all sexes, life expectancy in good health falls to around 65 years, almost three years after the legal retirement age (62 years). From a physical health standpoint, the transition to retirement is associated with increased risk factors such as unhealthy nutritional intakes [14], increased body mass and cholesterol levels [15] or even sedentary behavior [16]. In terms of psychological health, retirement can also be linked to more depressive symptoms [17] and cognitive decline. The results of three large-scale longitudinal surveys conducted in Europe and the United States point to a negative effect of retirement on cognitive performance. Recent research, however, qualifies this relationship by suggesting the occurrence of a slight decline [18,19] or by finding both a protective and a detrimental effect on cognition, depending, for example, on the type of occupation [20].

These public health issues must be combined with social and economic ones. It is estimated that by 2050, almost one-third of the population in France will be 60 years old, compared to only one-quarter in 2018 [21]. This number corresponds to an increase in the number of pensioners of about 1.3 per cent per year (DREES, 2020). A similar increase is occurring all around the world (OECD report) and generating societal problems, such as the costs associated with it [22]. Thus, retirement could also be seen as an economic burden for society [23]. These problems, which are widely publicized in the media, give rise to increasingly negative representations of ageing adults over time [24]. In France in particular, older adults are represented as less competent, in poorer health or dependent [23]. These representations should be considered, because they have significant effects on long-term health. In particular, the internalization of negative representations of ageing is associated with a reduction of nearly 7.5 years in life expectancy [25]. In contrast, more positive representations are linked to positive effects on health, including cognitive health [25]. Recently, a study conducted on a sample of 2253 people found that a higher subjective age was associated with accelerated epigenetic ageing. In other words, individuals who felt older than their chronological age were biologically older. This effect is thought to depend in part on perceived health and on inflammatory profiles [26]. These effects on health can be explained by behavioral changes due to representations of ageing. For example, representations of ageing could modify physical activity practices, as well as engagement in active health behaviors. These modifications could translate into physical changes, impacting long-term health (including cognition, [27]). However, it is difficult to identify the period in an individual’s life at which their effect is the most important, to limit the impact of these representations. One of the major stages of ageing that can coincide with this period is retirement [28]. Indeed, the retirement stage constitutes a major life crisis for individuals, associated with many changes occurring simultaneously [6,29]. Like age, it marks a change in social status, moving an individual from the category of active adults to that of older adults and retired people [30,31]. The context of retirement should then be seen as particularly relevant to activating and rendering self-relevant the representations of ageing [27]. Beyond this representation, retirement itself could be associated with specific representations of retirement.

A relative independence between representations of ageing and retirement is expected, as the older the individuals, the more likely they are to be identified with the group of older adults [32], but when they retire, individuals may also be less identified with the older adults group due to their younger age. However, they would be more likely to identify themselves with the retired group and thus be more vulnerable to representations of retirement that are directly relevant for them. Accordingly, some works have found that retirement is linked to specific representations such as boredom, loneliness and isolation and, somewhat less frequently, with illness and death [5].

To measure representations of retirement and to study their potential effects on health, Atchley validated one of the few existing scales: the Attitudes Toward Retirement Scale (1974). This is a semantic differential [33] composed of 16 pairs of bipolar adjectives, one considered negative and the other positive. The advantage of this method is to ensure that the attitude evaluation focuses on a particular object, in this case retired people. It was subsequently adapted and reduced to 14 items along two factors [1]: *mental health* related to coping problems (e.g., worthlessness or loneliness) and *physical health* related to physical health outcomes (e.g., lack of activity or illness).

This scale is useful for studying the long-term health effects of representations of retirement. The results of a study conducted with 394 American participants, followed over 23 years, showed that positive representations of retirement were associated with a survival advantage of almost five years, in contrast to people influenced by negative representations [1]. In addition, the results of a study of 1011 American participants from the Ohio Longitudinal Study of Ageing and Retirement (OLSAR) cohort, followed over 23 years, show firstly that a significant number of individuals expressed negative representations of retirement, with 76 per cent of participants expressing negative representations of physical health and 77 per cent expressing negative representations of mental health. Secondly, they found a survival advantage of four and a half years when the individual expressed positive representations of retirement concerning physical health and two and a half years concerning mental health [4].

This kind of scale thus appears relevant to study representations of retirement beyond those of ageing. Nonetheless, this scale has not yet been adapted into French, and no other similar tools are available. The objective of the present study is therefore to adapt and to validate for a French population this scale on representations of retirement. We chose the version of Lakra et al. [1], as this version is more recent and in line with the current global socio-economic and political context regarding retirement. We propose to rename this scale the *Retirement Semantic Differential* (RSD), for two reasons. First, the term ‘attitude’ is mainly used in relation to retirement to assess attitudes toward retirement planning and not social representations [34]. Secondly, the RSD is a semantic differential similar to a well-known scale in the field of ageism for assessing representations of ageing, the “Ageing Semantic Differential” [35].

## 2. Method

### 2.1. Translation and Adaptation

The original version from Lakra et al. [1] was firstly translated term by term by 2 researchers and a psychologist separately and then jointly. All of them are experts in cognitive and social ageing and ageism and have some expertise in English and French. The translated version was then culturally adapted and unified after a consensual meeting. Since the questionnaire is not based on sentence proposals, no translation/retranslation was carried out. The French adaptation is presented in Table 1, along with the original English version.

However, we felt it necessary to adapt the scale assessing representations of retirement for cultural specificity, here French. Retirement is a societal construct, and many differences can be observed in the existing practices and representations around retirement in each country. Indeed, cultural differences depend on public policies and the history of the country in terms of labor and retirement laws [36].

### 2.2. Participants

The total sample consisted of 279 participants, recruited online, who had to be between 18 and 55 years old (*M_age_* = 33.5 years, *SD_age_* = 12), of which 15.5 per cent were men (*N* = 45) and 84.5 per cent were women (*N* = 234). The mean education level of the sample was of 14.8 years ± 2.31. Participants were from all socio-economic and socio-cultural backgrounds. They were representative of the general population. Participants over 55 years of age were excluded from the sample, to prevent their imminent proximity to retirement from specifically influencing their responses. Being closer to this major transition, their representations of retirement might be more based on imminent personal experiences rather than on general perceptions. This exclusion thus aims to ensure that the scale measures representations of retirement among individuals for whom this stage is still relatively distant, thereby ensuring the better validity of the results for the target population. Moreover, with regard to representations of ageing, older people could show a change in their representations compared to younger people [37]. A similar effect on people approaching retirement could be assumed for representations of retirement, and the objective of the present study is to adapt and validate a scale comparable to that proposed by Lakra et al. [1], for a similar population.

The sample size follows the recommendations used for the adaptation and validation of a scale, i.e., a minimum of ten participants per item [38], as the scale is composed of 14 items. This study was approved by the university ethics committee, and non-opposition for everyone was collected at the very beginning of the survey in accordance with French law.

### 2.3. Material

The RSD scale is composed of 14 items; for each item, participants were asked to categorize retired people according to a pair of opposing adjectives such as dependent/independent with a seven-point Likert scale, 1 being assigned to more positive adjectives such as ‘independent’ and 7 to more negative adjectives such as ‘dependent’. A total score (out of 98) is obtained, and the higher the score, the more negative the representations of retirement, and conversely, the lower the score, the more positive they are. An average score corresponds to neutral representations of retirement.

### 2.4. Procedure

Participants responded online, using a form developed with LimeSurvey (version 3.24.3 + 201027), hosted on the laboratory’s servers. The dissemination of the study was mainly carried out on social networks (Facebook and LinkedIn) and via the mailing lists of local associations. No specific individuals were targeted for recruitment. Participants were recruited from all over France.

### 2.5. Statistical Analysis and Validity

The total sample (*N* = 279) was used to conduct an exploratory factor analysis (EFA). In addition, a construct validity analysis and an analysis of the effects of differences between sexes, age and education on the RSD scores were also conducted on the total sample. A value of *p* < 0.05 was used as a significance threshold. All analyses were conducted using Jamovi software, version 1.1.9.0 (the Jamovi project, 2020) and R software (4.0.2) (R Development Core Team, 2021) (to compute a Mardia test that could not be computed with Jamovi).

### 2.6. Exploratory Factor Analysis

The Kaiser–Meyer–Olkin (KMO) test and Bartlett’s test of sphericity were calculated on the total sample to measure the sample adequacy required for EFA. For the KMO, the lowest acceptable limit was 0.50 [39]. For the Bartlett test, a significant value (*p* < 0.05) indicates that the variables are sufficiently related. To assess the probability that this sample is drawn from a multinormal population, we used the Mardia test [40]. If this test is significant, the data deviate from normality. It is then recommended to use the principal axis factor rather than the maximum likelihood as an appropriate estimator [41]. A parallel analysis was performed to indicate the best factorial solution to be used. The EFA was conducted with oblique rotation, as it is assumed that the factors are correlated with each other. A loading of 0.40 in one factor and a difference of 0.30 with the loadings on the other factors were used as a cut-off [41].

### 2.7. Reliability of Constructs

The reliability of the constructs was tested by calculating the standard factor load (SLF), the composite reliability (CR) and the average of the extracted variances (AVE). Concerning the SLF, the threshold of saturation considered for each item on its scale was 0.50. The CR index was calculated using Cronbach’s Alpha and McDonald’s Omega, considering a threshold of acceptability higher than 0.70. Finally, the AVE had to be greater than 0.50 for each item [42].

### 2.8. Difference between Sexes, Age, Education and RSD Score

The possible effect of sex (between subject variables) on the RSC scores (total and by factor) was assessed by analyses of variance (ANOVAs). The relationship between age and education and the RSD scores (total and by factor) was evaluated through linear regression analyses. Beforehand, the different conditions for applying an ANOVA and a linear regression were checked: the normality of the distributions as assessed by the skewness and kurtosis index <|2| and the homogeneity of the variances and the absence of outliers or extreme values. An analysis of absolute deviation from the median (MAD method, [43]), was performed by subgroup on each variable of interest (RSD, age and education), to detect the presence of extreme values (threshold = 3). Following this procedure, two individuals were excluded for the RSD variable.

## 3. Results

### 3.1. Exploratory Factor Analysis

A KMO value of 0.91 is observed, indicating that the correlation patterns are compact and that the factor analysis should produce distinct and reliable factors, as do the KMO values for each individual item (all >0.80; threshold at 0.50) (see Table 2). The Bartlett’s test of sphericity was significant χ^2^ (91) = 1729; *p* < 0.001, which means that the correlation matrix is suitable.

The Mardia test was statistically significant for skewness and kurtosis (*Mardia skewness* = 32.13, *χ*^2^(*N* = 217) = 1161.92, *p* < 0.05; *Mardia kurtosis* = 276.77, *Z* = 18.36, *p* < 0.000); then, the principal axis factor extraction method was chosen.

The parallel analysis suggested a three-factor model. In accordance with the item inclusion criteria, for factor one, item three (uniqueness < 0.40) was removed. For factor two, item two (uniqueness < 0.40) and items four (uniqueness < 0.40) were also excluded. No items were deleted from factor three (see Table 3).

Thus, the final scale consists of 11 items divided into three factors. Factor one consists of four items (hopeless/hopeful; worthless/worthy; dissatisfied/satisfied; empty/full) consistent with the “mental health” factor described by Lakra et al. [1]. The second factor consists of five items (sick/healthy; immobile/mobile; involved/uninvolved; dependent/independent; idle/busy) which is consistent with the “physical health” factor found by [1], and the “Physical Potency” and “Activity” factors reported by Atchley [44]. Thus, similarly, we have named it “physical health”. Finally, the third factor includes two items (unable/able; meaningless/meaningful) not previously identified as belonging to a similar factor. We identified it as “value” (see Table 3).

### 3.2. Reliability of Constructs

The SFL was above 0.50 for all items except item 14 (SFL = 0.42), which was at the limit [45]. As for the CR indices, they appear acceptable (*α* = 0.899 and Ω = 0.901). Similarly, the AVE for each item was above the threshold (see Table 3).

### 3.3. Difference between Sexe, Age, Education and RSD Score

There was no significant effect of the sexes on the overall RSD score, with *F* (1.275) = 1.13, *p* = 0.288. Concerning the individual factors, the results show no effect of the sexes for both factor one, “mental health” *F* (1.275) = 0.005, *p* = 0.94, and for factor two, “physical health” *F* (1.275) = 1.72, *p* = 0.19. However, a sex effect is observed for factor three, “value”, with *F* (1.275) = 4.60, *p* = 0.03, *η^2^* = 0.02. Women have a lower score on the RSD scale than men (women = 5.96 ± 0.16; men = 7.07 ± 0.35), meaning that they express more positive (or less negative) representations of retirement.

Concerning education and age effects, there was no effect of education *F* (1.275) = 0.731, *p* = 0.393. However, there was a significant effect of age on the overall RSD score, with *F* (1.275) = 6.84, *p* = 0.009, *R*^2^ = 0.03. The higher the age of the participants, the lower the RSD score (see Figure 1).

For individual factors, the results show no effect of education for factor one, “mental health”, *F* (1.275) = 0.425, *p* = 0.515, factor two, “physical health” *F* (1.275) = 1.82, *p* = 0.178, and factor three, “value” *F* (1.275) = 0.06, *p* = 0.796. In addition, there was a difference between age and the scores for factors 1 and 2, respectively: *F* (1.275) = 8.39, *p* = 0.004, *R*^2^ = 0.03, and *F* (1.275) = 4.80, *p* = 0.029, *R*^2^ = 0.02. As with the RSD total score, the higher the age of the participants, the lower the factor 1 and factor 2 scores. But there was no effect of age for factor three, “value”, *F* (1.275) = 0.518, *p* = 0.472.

## 4. Discussions

The objective of this study was to adapt and to validate the Retirement Semantic Differential (RSD) scale in a French population. This scale measures social representations of retired people and would be the first scale of this kind available in French. Like representations of ageing, these representations may have significant effects on long-term health, particularly in the context of retirement. It was therefore important to be able to assess them and differentiate them from representations of ageing to prevent their effects and promote healthy ageing.

The results showed that the RSD scale has good psychometric properties. The AFE found a reduced scale structure of 11 items (compared to 14 previously, [1]), divided into three factors: *mental health*, *physical health* and *value* (Appendix A). The EFA was found to be relevant in comparison to a confirmatory factor analysis because retirement is a societal construct, and many differences can be observed in the existing practices and representations around retirement in each country [36]. This adaptation remains consistent with the factor structure found by Lakra et al. [1], while some minor differences were found. These differences concern two items excluded from factor two, “physical health” (compared to Lakra et al.), and the addition of a new factor three, “value”, comprising two items (“uninvolved/involved” and “idle/busy”). Furthermore, the factorial distribution analysis identified distinct effects of age and sex on the total and factor scores. It was observed that age had a negative effect on the total RSD score, as well as on the first two factors, “mental health” and “physical health”. The score for the third factor, “value”, does not seem to be affected by the age of the participants.

To our knowledge, this age effect had never been tested in relation to retirement representations. However, a similar effect is observed in the literature on representations of ageing. Indeed, the older people are, the less negative their representations of ageing are [46]. This effect can be explained by a process of accommodation and assimilation used to cope with age-related changes and to protect oneself from the deleterious effects of negative representations of ageing on mental health [47]. Moreover, like representations of ageing, the age of the participants has an impact on the representations of retirement in relation to health (factors 1 and 2). One of the explanations for this observation is based on the hypothesis that the increased medicalization of ageing in recent years tends to increase the negativity of representations of ageing and their link with health [24]. Since retirement is associated with ageing, it seems that a similar effect is observed in the health-related domains concerning representations of retirement. However, these results on the effects of age on social representations also limit the generalization of our scale and its use for an older population. It seems essential that future studies assess representations of age and retirement in an older population. It can be assumed, however, that due to the societal and cultural character of representations, only the valence of representations changes with age and not the structure of representations (see [48] for a detailed method of studying social representations).

Moreover, factor three, “value”, not being related to health, does not seem to be sensitive to the age of the participants. However, unlike age, the sexes have an effect on this factor, with women showing fewer negative representations than men. This effect may be due to the different distribution of men and women in the sample of participants, and it appears difficult to establish a conclusion on this effect in these conditions. In fact, only 15.5 per cent of the participants were men. Thus, future studies could re-examine this gender effect on retirement representations with a larger sample of men. In addition, this effect can also be explained by an effect of sex representations. Indeed, men are more generally associated with the terms included in factor three, such as “competent” [49]; the retirement stage, with the cessation of professional activity, would generate more negative representations for them on this theme. Social identity and role expectations may play a role; men often derive a significant portion of their identity and self-worth from their professional roles due to societal expectations. Retirement may lead to a perceived loss of status and purpose, resulting in more negative representations of value. Women, who traditionally balance multiple roles—professional, familial, and social—might experience a smoother transition, maintaining a sense of value beyond their careers. Therefore, these social factors could contribute to the observed gender differences on the “value” factor.

Regarding the factorial distribution and the deletion of some items, these changes can be explained by cultural and translation differences. First, three items were deleted from this adaptation, because they were not sufficiently related to the others: “enthusiastic/angry”, “active/inactive” and “happy/sad”. The deletion of the item “active/inactive” could be explained by a difference in translation. The terms “active” or “inactive” in French are used to define professional activity or inactivity, so it appears contradictory for the participant to categorize a retired person as “active” (or “inactive”), as the terms are almost synonymous and antonymous. In comparison, the terms “active” and “inactive” in English are less systematically associated with professional status, meaning rather “dynamic”.

With regard to the other two deleted items, personality theory, traits or state could help explain this exclusion (see [50]). Indeed, these two items seem to refer to an individual emotional reaction about specific situations, like a transitory state in response to a positive or negative event. In contrast, the other items seem to be less dependent on a specific event, but more about long-term changes, such as changes in activities performed at the time of retirement (e.g., idle vs. busy). Thus, while the other items seem to define “traits” that are less sensitive to specific short-term events, the excluded items refer to more transient “states” [51]. The RSD scale appears to assess attitudes toward retirees that are oriented toward long-term change or evolution, which is consistent with the period of major change that retirement represents in an individual’s life [52].

Secondly, two items, initially present in the *mental health* factor, are found in this French adaptation in the *physical health* factor: the items “uninvolved/involved” and “idle/busy”. These two items refer to an idea of adaptation and activity after retirement (see [4]). In the initial version of the RSD scale validated in an American population, the activity of retirees seems to be dependent on mental health, but for a French population it seems to be dependent on physical health. Although few studies report cultural differences in the perception of overall health between France and the United States, an exploratory study shows that for Americans, health seems to be represented more by psycho-affective aspects, and for Europeans it appears to be represented more by physical autonomy and independence [53]. Furthermore, it appears that cultural differences between North America and Europe in the occupation of time after retirement may modulate representations of ageing and their valence (see [29] for a Belgian/Canadian comparison). Similarly, results suggest that the active participation of older people in voluntary activities leads to a greater perception of their competence independently of their actual competence [54]. Moreover, social representations are dependent on the surrounding culture (see [55]), and the cultural differences between an American population (RSD validation in English, Lakra et al. [1]) and a French population can explain the new factorial distribution. Although not yet studied directly in regard to retirement representations, it can be assumed that culture and public policies also play a significant role in the content of retirement representations and their valence.

The current socio-political context in Western societies, and France in particular, could also contribute to understanding this third factor, “value”. This factor seems to capture the abilities of retired people to be competent and important, and thus their social value. From a societal point of view, the increased number of newly retired people could lead to a forecast of a major shortage of workers around the world, as in Europe [22]. The unbalanced relationship between the number of retirees and the active labor force, as well as the increase in expenditure, generates attitudes, generally negative, toward retirees, who are seen as “burdensome” to the economy and society [5]. Moreover, retirement is generally associated with ageing [30], and opinions about older individuals are largely negative in our society. The ageing person is generally associated with less activity, linked to negative representations such as illness, slowness or more generally difficulties [56]. In this sense, the retired person would represent less “value” for society in terms of work capacity and economic usefulness. This type of representation is also observed in the workplace. Older workers suffer from negative representations; they are generally perceived as less competent and less productive than younger workers [57]. The third factor, “value”, may thus come from the current socio-political context, in which the political and media debates concerning retirement have intensified considerably over the last ten years. This is particularly true in France since the announcement of the pension reform planned for 2022. This evolution in France can be observed in many other countries, so it can be assumed that in the next few years, this third factor questioning the representations of the “social value” of retired people may become generalized.

This adaptation also has some limits, such as the lack of assessment of the absence of the temporal stability of the RSD as well as the subsequent need for a confirmatory analysis. From a theoretical point of view, representations as defined by Abric (1987), then Flament (1994), are a set made up of a central core of representations which have the particularity of being common to all, stable in time and fluctuating little according to the context. Around this central core is the peripheral system made up of more fluctuating representations, which can vary according to the context and the life history of the individual. For example, it seems that the peripheral system and even to a lesser extent the central core of representations of retirement may be sensitive to the age of the participants [5]. Thus, in the context of this scale adaptation, it will be relevant for future studies to carry out a construct stability analysis and a confirmatory analysis, to be able to identify more precisely the central or peripheral representations. However, it is important to note that this translation did not follow all the recommended steps for intercultural validation. Future studies using this scale should follow these guidelines [58], to ensure a more rigorous validation process.

Furthermore, future research should aim to study not only the impact of retirement representations on health after retirement, but also their effects upstream of retirement, in the pre-retirement period. The temporal process model of retirement postulates that the retirement process involves three sequential phases: retirement planning, retirement decision-making, transition to retirement and adjustment [59]. The first two phases (i.e., retirement planning and retirement decision-making) are crucial for predicting the adjustment to retirement (see [60]. However, representations of retirement are likely to have an impact on these first two phases as well, like pre-retirement, and not only on the transition to retirement (post-retirement), as has already been studied. It is even assumed that the retirement stage could act as a catalyst for the effect of representations, which could partly explain the repercussions of the transition to retirement on health [27,61]. Preparing for the transition to retirement appears to promote a favourable evolution of health [30]. Conversely, a lack of preparation is associated with more negativity (see [62]). Thus, to limit the harmful effects of negative representations of retirement on health, it seems essential to be able to propose pre-retirement interventions by integrating the awareness of representations of retirement into the preparation for retirement. The use of the RSD scale and the various dimensions it identifies will make it possible to target the proposed interventions more effectively.

In conclusion, to our knowledge, this is the first adaptation and validation in French of a scale for measuring representations of retirement. The results led to a reliable and valid version of the Retirement Semantic Differential (RSD) scale reduced to 11 items divided into 3 factors. This factorial distribution shows that attitudes toward retired people can be distinguished in several dimensions consistent with the current socio-political context in terms of retirement. Moreover, the RSD scale seems relevant for evaluating the evolution of representations and their effects at different times of retirement: pre- and post-retirement. This should subsequently allow the development of intervention and retirement-planning programs that promote healthy ageing.

## Figures and Tables

**Figure 1 behavsci-14-00891-f001:**
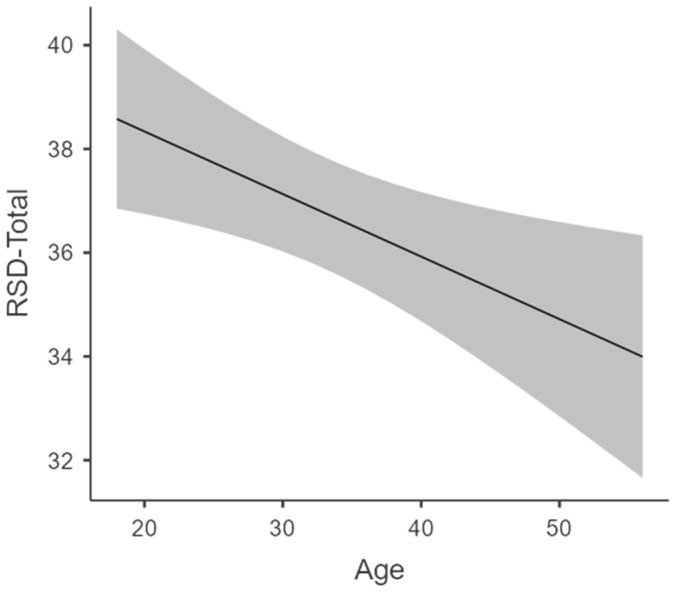
Relationship between age and RSD total score. Gray ribbons represent the 95% confidence interval and the dark line the estimated marginal means.

**Table 1 behavsci-14-00891-t001:** Item translation from English to French.

RSD Items	English	French
Item 1	Sick/Healthy	Malade/En bonne santé
Item 2	Bad/Good	Aigrie/Enthousiaste
Item 3	Inactive/Active	Inactive/Active
Item 4	Sad/Happy	Triste/Heureuse
Item 5	Immobile/Mobile	Immobile/Mobile
Item 6	Uninvolved/Involved	En retrait/Engagée
Item 7	Unable/Able	Inapte/Compétente
Item 8	Dependent/Independent	Dépendante/Indépendante
Item 9	Hopeless/Hopeful	Pessimiste/Optimiste
Item 10	Worthless/Worthy	Abjecte/Digne
Item 11	Dissatisfied/Satisfied	Insatisfaite/Satisfaite
Item 12	Empty/Full	Renfrognée/Épanouie|
Item 13	Idle/Busy	Oisive/Occupée
Item 14	Meaningless/Meaningful	Anodine/Importante

**Table 2 behavsci-14-00891-t002:** Analysis of dimensionality and convergence validity of Retirement Semantic Differential (RSD).

RSD Items	KMO	Component Loading	Cronbach’s Alpha	McDonald’s Omega
Global	0.91		0.897	0.901
Item 1 Sick/Healthy	0.930	0.668	0.894	0.898
Item 2 Bad/Good	0.930	0.394 +	0.885	0.889
Item 3 Inactive/Active	0.885	0.479 +	0.886	0.892
Item 4 Sad/Happy	0.900	0.405 +	0.886	0.889
Item 5 Immobile/Mobile	0.931	0.519	0.888	0.893
Item 6 Uninvolved/Involved	0.912	0.628	0.892	0.896
Item 7 Unable/Able	0.914	0.545	0.889	0.894
Item 8 Dependent/Independent	0.917	0.552	0.890	0.894
Item 9 Hopeless/Hopeful	0.932	0.560	0.890	0.894
Item 10 Worthless/Worthy	0.937	0.620	0.891	0.896
Item 11 Dissatisfied/Satisfied	0.934	0.520	0.889	0.893
Item 12 Empty/Full	0.909	0.420	0.886	0.890
Item 13 Idle/Busy	0.887	0.686	0.894	0.898
Item 14 Meaningless/Meaningful	0.769	0.830	0.901	0.903

Note: + no-significance level.

**Table 3 behavsci-14-00891-t003:** Factorial distribution.

RSD Items	Factor 1 Mental Health	Factor 2 Physical Health	Factor 3 Value	Uniqueness
Item 1 Sick/Healthy		0.547		0.592
Item 2 Bad/Good	0.795			0.340 +
Item 3 Inactive/Active		0.863		0.274 +
Item 4 Sad/Happy	0.799			0.354 +
Item 5 Immobile/Mobile		0.707		0.452
Item 6 Uninvolved/Involved		0.390		0.664
Item 7 Unable/Able			0.357	0.536
Item 8 Dependent/Independent		0.507		0.508
Item 9 Hopeless/Hopeful	0.710			0.520
Item 10 Worthless/Worthy	0.353			0.663
Item 11 Dissatisfied/Satisfied	0.776			0.459
Item 12 Empty/Full	0.640			0.450
Item 13 Idle/Busy		0.440		0.688
Item 14 Meaningless/Meaningful			0.733	0.427

Note: + no-significance level.

## Data Availability

The datasets used and/or analyzed during the current study are available from the corresponding author on reasonable request.

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
