# Peer review of "A French Adaptation and Validation of Retirement Semantic Differential (RSD)"

_behavsci, 2024, doi:10.3390/bs14100891_

Round 1

Reviewer 1 Report

Comments and Suggestions for Authors

This is a carefully done study, and the findings are of considerable interest. A few minor revisions are listed below.

Possible additional reasons for limiting the age of participants to between 18 and 55 years old.

In the methods or results sections, there is less explanation of what a high or low result in RSD scores means.

In the discussion section, the effect of differences in the gender ratio of the participants on the results of the study is less explained.Reasonable explanations could be added appropriately.

Author Response

Comments 1 : Possible additional reasons for limiting the age of participants to between 18 and 55 years old.

Response : Thank you for your comment, we have added this explanation line 180 :"Participants over 55 years of age were excluded from the sample because, to prevent their imminent proximity to retirement from specifically influencing their responses. Being closer to this major transition, their representations of retirement might be more based on imminent personal experiences rather than on general perceptions. This exclusion thus aims to ensure that the scale measures representations of retirement among individuals for whom this stage is still relatively distant, thereby ensuring better validity of the results for the target population. Moreover, with regard to representations of ageing, older people would show a change in their representations compared to younger people (see Hess et al., 2009). "

Comments 2 : In the methods or results sections, there is less explanation of what a high or low result in RSD scores means.

Response : we've changed the explanation on line 201 : "The total score (out of 98) is obtained, and the higher the score, the more negative the representations of retirement, and conversely, the lower the score, the more positive they are. An average score corresponds to neutral representations of retirement."

But, we cannot establish a definitive threshold for the negativity or positivity of representations, as interpretations are always relative to context.

Comments 3 : In the discussion section, the effect of differences in the gender ratio of the participants on the results of the study is less explained.Reasonable explanations could be added appropriately.

Response : We have add an other justification ligne 355 : "Indeed, men are more generally associated with the terms included in factor three, such as "competent" (see Yzerbyt et al., 2021), the retirement stage, with the cessation of professional activity, would generate more negative representations of them on this theme. Social identity and role expectations may play a role; men often derive a significant portion of their identity and self-worth from their professional roles due to societal expectations. Retirement may lead to a perceived loss of status and purpose, resulting in more negative representations of value. Women, who traditionally balance multiple roles—professional, familial, and social—might experience a smoother transition, maintaining a sense of value beyond their careers. Therefore, these social factors could contribute to the observed gender differences on the "value" factor."

Reviewer 2 Report

Comments and Suggestions for Authors

Summary:

The aim of this study was to take an existing measure of retirement perceptions, translate it into French, and test the psychometric properties using a French sample.

The original measure was a semantic differential scale composed of 16 pairs of adjectives that are opposite one another. The original measure was published by Osgood et al. (1957) and later validated by Atchley (1974). Then, it was tested and validated by Lakra et al. (2012), and the present investigation used the Lakra et al. (2012) version.

Participants in the present investigation were 279 French adults between the ages of 18 and 55 and they completed the scale online. The authors performed an exploratory factor analysis and a reliability analysis with a priori cut-off values. The exploratory factor analysis suggested a 3-factor model, in contrast to the previously established 2-factor model. Items 2, 3, and 4 were removed in accordance with the established inclusion criteria. For the current version of the scale, factor 1 is mental health, factor 2 is physical health, and the (new) third factor is value. Reliability analyses indicated that all scale items had acceptable reliability and Cronbach’s alpha for the overall scale was good (0.901). Additional analyses were performed to examine the effects of sex, age, and education. The authors found that women have a more positive representation of retirement than men and that retirement perceptions became more positive with increased age. Results are discussed in terms of how French culture and public policies may have contributed to the new factor structure of the measure.

Comments:

1. This article is on an important topic. As the authors discuss in the literature review, the population is aging and it is worthwhile to examine retirement perceptions.

2. This article is strongly grounded in previous research (studies on the same measure done by Atchley and Lakra et al.).

3. During my cursory scan of the article, I thought that the RSD had a dichotomous response format where the participant picked the adjective that they thought represented retired people more. But upon a careful read, I realized that the items were on a 7-point Likert-type scale and that the positive and negative adjectives were the anchor terms. The authors deleted 3 items from the scale, but I don’t think that a final version of the 11-item new scale is in any of the tables. For these reasons, would it be possible to include an appendix with the final 11-item version of the scale showing the correct response format? For example, item 1 would be:

Healthy

Sick

1

2

3

4

5

6

7

4. If I understand correctly, lower scores represent more positive representations of retirement and higher scores indicate more negative representations. This seems counter-intuitive. Is it possible to reverse the direction? Was this direction chosen to keep consistent with previously published versions of the scale?

5. One of the analyses performed was to examine sex / gender differences, but the breakdown of male and female participants was very uneven (84.5 percent women compared to 15.5 percent men). Were statistical tests done to examine whether the variances were homogenous? If not, were corrections applied?

6. In section 3.3 “Difference Between Sexes, Age, Education, and RSD Score,” I am a little bit confused on these sentences:

“There was no significant effect of the sexes on the overall RSD score with F (1.275) = 276 1.13, p = 0.288. Concerning the individual factors, the results show no effect of the sexes 277 for both factor one "mental health" F (1.275) = 0.005, p = 0.94 and for factor two "physical 278 health" F (1.275) = 1.72, p = 0.19. However, a sex effect is only observed for this factor with 279 F (1.275) = 4.60, p = 0.03, η² = 0.02. Women have a lower score on the RSD scale than men 280 (women = 5.96 ± 0.16; men = 7.07 ± 0.35), meaning that they have more positive (or less 281 negative) representations of retirement.”

When the authors write, “…a sex effect is only observed for this factor….” Does that refer to factor 2? Or factor 3?

Comments on the Quality of English Language

There are minor typing mistakes.

Author Response

1. Summary

2. Point-by-point response to Comments and Suggestions for Authors

Comments 3: During my cursory scan of the article, I thought that the RSD had a dichotomous response format where the participant picked the adjective that they thought represented retired people more. But upon a careful read, I realized that the items were on a 7-point Likert-type scale and that the positive and negative adjectives were the anchor terms. The authors deleted 3 items from the scale, but I don’t think that a final version of the 11-item new scale is in any of the tables. For these reasons, would it be possible to include an appendix with the final 11-item version of the scale showing the correct response format? 

Response: Thank you for your comment, it could make it easier to use the scale later on, so we've added an appendix with the final version of the RSD in French. (line 489)

Comments 4: If I understand correctly, lower scores represent more positive representations of retirement and higher scores indicate more negative representations. This seems counter-intuitive. Is it possible to reverse the direction? Was this direction chosen to keep consistent with previously published versions of the scale?

Response: Thank you for your feedback. The direction of the scale is indeed consistent with the later scale for measuring representations of retirement. What's more, this direction is consistent with most scales measuring representations, such as age. So, to facilitate the use of this scale in connection with other similar scales, we wish to retain this meaning.

Comments 5: One of the analyses performed was to examine sex / gender differences, but the breakdown of male and female participants was very uneven (84.5 percent women compared to 15.5 percent men). Were statistical tests done to examine whether the variances were homogenous? If not, were corrections applied?

Response: Indeed, this is a valid point. We conducted a Levene's test for homogeneity of variances, which yielded a non-significant result (p = 0.322).

Comments 6: In section 3.3 “Difference Between Sexes, Age, Education, and RSD Score,” I am a little bit confused on these sentences:

“There was no significant effect of the sexes on the overall RSD score with F (1.275) = 276 1.13, p = 0.288. Concerning the individual factors, the results show no effect of the sexes 277 for both factor one "mental health" F (1.275) = 0.005, p = 0.94 and for factor two "physical 278 health" F (1.275) = 1.72, p = 0.19. However, a sex effect is only observed for this factor with 279 F (1.275) = 4.60, p = 0.03, η² = 0.02. Women have a lower score on the RSD scale than men 280 (women = 5.96 ± 0.16; men = 7.07 ± 0.35), meaning that they have more positive (or less 281 negative) representations of retirement.”

When the authors write, “…a sex effect is only observed for this factor….” Does that refer to factor 2? Or factor 3?

Response: Thank you, it was a mistake, here is a corrected version (line 282):  “There was no significant effect of the sexes on the overall RSD score with F (1.275) = 1.13, p = 0.288. Concerning the individual factors, the results show no effect of the sexes for both factor one "mental health" F (1.275) = 0.005, p = 0.94 and for factor two "physical health" F (1.275) = 1.72, p = 0.19. However, a sex effect is only observed the factor three “value” with F (1.275) = 4.60, p = 0.03, η² = 0.02. Women have a lower score on the RSD scale than men (women = 5.96 ± 0.16; men = 7.07 ± 0.35), meaning that they have more positive (or less negative) representations of retirement.”